# Persona Alchemy: Designing, Evaluating, and Implementing Psychologically-Grounded LLM Agents for Diverse Stakeholder Representation

**Sola Kim**
School of Sustainability
Arizona State University
Tempe, AZ 85281, USA
{sola}@asu.edu

**Dongjune Chang**
Applied Materials Division
Argonne National Laboratory
Lemont, IL 60439, USA
{changd}@anl.gov

**Jieshu Wang**
Department of Technology and Society
Stony Brook, NY 11794, USA
{jieshu.wang}@stonybrook.edu

## Abstract

Despite advances in designing personas for Large Language Models (LLM), challenges remain in aligning them with human cognitive processes and representing diverse stakeholder perspectives. We introduce a Social Cognitive Theory (SCT) agent design framework for designing, evaluating, and implementing psychologically grounded LLMs with consistent behavior. Our framework operationalizes SCT through four personal factors (cognitive, motivational, biological, and affective) for designing, six quantifiable constructs for evaluating, and a graph database-backed architecture for implementing stakeholder personas. Experiments tested agents' responses to contradicting information of varying reliability. In the highly polarized renewable energy transition discourse, we design five diverse agents with distinct ideologies, roles, and stakes to examine stakeholder representation. The evaluation of these agents in contradictory scenarios occurs through comprehensive processes that implement the SCT. Results show consistent response patterns ($R^2$ range: $0.58 - 0.61$) and systematic temporal development of SCT construct effects. Principal component analysis identifies two dimensions explaining 73% of variance, validating the theoretical structure. Our framework offers improved explainability and reproducibility compared to black-box approaches. This work contributes to ongoing efforts to improve diverse stakeholder representation while maintaining psychological consistency in LLM personas.

## 1 Introduction

Current LLM agents rely on static prompts and struggle to represent diverse stakeholder perspectives with psychological plausibility (Appendix A). Traditional personality theories provide limited insight into how behavior is shaped through interactions, and existing persona-conditioning methods lack consistency, adaptability, and representativeness. We propose a Social Cognitive Theory (SCT)

framework that grounds LLM agent personas in established psychological principles, enabling dynamic agents that evolve through interactions. Our contribution is threefold: we demonstrate how psychological theories inform LLM design and evaluation; we provide evidence that interdisciplinary design principles yield measurable outcomes; and we offer a reproducible methodology for designing and evaluating LLM personas in transdisciplinary domains such as sustainability.

## 2 SCT-BASED AGENT FRAMEWORK

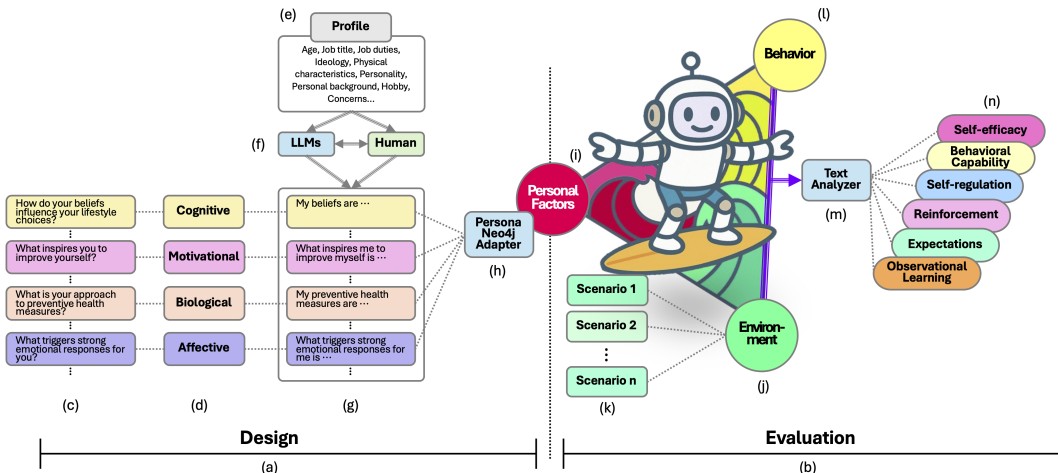

Figure 1: SCT Framework Using Personal Factors for Agent Design and Six Constructs for Cross-Scenario Evaluation within Triadic Reciprocal Determinism. *Note:* Light blue round squares indicate LLMs in the framework.

Our framework (Figure 1) is grounded in SCT's "triadic reciprocal determinism" (Bandura, 2023), which models the continuous interaction among personal factors, environment, and behavior. We operationalize SCT through four personal factors—cognitive, motivational, biological, and affective—implemented as a 550-question dataset forming each agent's persona foundation (Appendix B; Appendix F). Six SCT constructs (self-efficacy, behavioral capability, expectations, self-regulation, observational learning, and reinforcements) serve as quantifiable evaluation metrics assessing persona consistency when agents face contradicting information (Appendix C). The architecture uses a Neo4j graph database (Neo4j, Inc, 2025) with `Llama-3.2-3B-Instruct` (Meta AI, 2024) as the base model, enabling hierarchical persona data retrieval via semantic similarity during conversations (Appendix G; Appendix H).

## 3 CASE STUDY AND EXPERIMENTAL DESIGN

We tested our framework on renewable energy transition discourse, a domain characterized by cultural polarization and conflicting stakeholder interests (Kahan et al., 2015; Ruggiero et al., 2014). Five diverse agent profiles with distinct ideologies were developed (Figure 3; Appendix D), each facing contradictory information scenarios with varying source reliability across 5 interaction rounds (100 iterations per condition). The agent architecture leverages a neuroscience-inspired Enhanced Memory System (Chang & Kim, 2025) with RAG-based retrieval and dynamic SCT construct tracking. Response patterns were modeled using two hierarchical linear models: a fixed-effects model (Model 1) and a time-varying model (Model 2), with a likelihood ratio test ($\Lambda = 399.82$, $p < .001$) confirming that temporal interactions improve model fit (Appendix I.1.3; Appendix J).

## 4 RESULTS

**Response patterns.** The fixed-effects model (Model 1) showed that contradicting information consistently predicted SCT-based response patterns across all agents, with stable coefficients ($\beta_1 =$

1.71–1.74) and high explanatory power ($R^2$: 0.58–0.61; Appendix E, Table 4). SCT-based agents demonstrated nearly 5-fold stronger responses compared to a vanilla agent ($\sim$1.73 vs. 0.36), whose higher $R^2$ (0.83) with lower coefficient suggests rigid, less psychologically plausible dynamics. Agent differences were statistically insignificant when controlling for contradictory information (all $p > .85$, $\eta^2 = 0.0002$), confirming framework robustness.

**Temporal development.** Table 1 presents the time-varying model (Model 2) parameters, showing statistically significant interactions ($p < .05$) between SCT constructs and interaction rounds. Self-efficacy shows the strongest positive trajectory ($\beta = 0.318$, $p = .002$), indicating increasing resistance to contradicting information. Observational learning also trends positively ($\beta = 0.115$, $p = .035$), while expectations ($\beta = -0.211$, $p < .001$), reinforcements ($\beta = -0.172$, $p = .025$), and self-regulation ($\beta = -0.135$, $p = .007$) show increasing responsiveness over time. Behavioral capability remains stable ($p = .387$). Figure 2b visualizes these trajectories (summary statistics in Appendix E, Table 5).

| Parameter | Coefficient | SE | p-value |
|---|---|---|---|
| Intercept | -0.127 | 0.038 | 0.999 |
| Source reliability | 1.426 | 0.020 | <0.001*** |
| *SCT Constructs* | | | |
| Self-efficacy | 2.510 | 0.220 | 0.999 |
| Behavioral capability | -0.124 | 0.151 | 0.999 |
| Expectations | -1.569 | 0.138 | 0.999 |
| Self-regulation | -1.373 | 0.114 | 0.999 |
| Observational learning | 1.397 | 0.110 | 0.999 |
| Reinforcements | -1.871 | 0.151 | 0.999 |
| *Temporal Development* | | | |
| Self-efficacy $\times$ round | 0.318 | 0.103 | 0.002** |
| Behavioral capability $\times$ round | -0.036 | 0.042 | 0.387 |
| Expectations $\times$ round | -0.211 | 0.059 | <0.001*** |
| Self-regulation $\times$ round | -0.135 | 0.050 | 0.007** |
| Observational learning $\times$ round | 0.115 | 0.055 | 0.035* |
| Reinforcements $\times$ round | -0.172 | 0.077 | 0.025* |
| Random effects variance | $7.36 \times 10^{-6}$ | – | 0.951 |
| Residual variance | 0.035 | – | – |

*Note*: $^{*}p < 0.05$, $^{**}p < 0.01$, $^{***}p < 0.001$

Table 1: Temporal Development Effects Model (Model 2) Parameter Estimates

**PCA validation.** Principal component analysis (Figure 2a) revealed two components explaining 73% of variance: PC1 (46%) captures a general "response tendency" with positive loadings across all constructs, while PC2 (27%) differentiates learning-oriented constructs (Observational Learning, Self-regulation) from expectation-based constructs, validating SCT's theoretical structure (Appendix E, Table 6).

## 5 CONCLUSION

Our SCT-based framework advances LLM persona design by providing psychologically grounded stakeholder representations that balance stability with adaptability. Results demonstrate consistent persona dynamics across diverse agents and systematic temporal development of SCT construct effects. The methodology offers a reproducible, explainable framework for evaluating persona consistency applicable beyond renewable energy. Future work integrating complementary psychological models and multi-agent interactions will enhance artificial social systems in education, healthcare, and sustainable development.

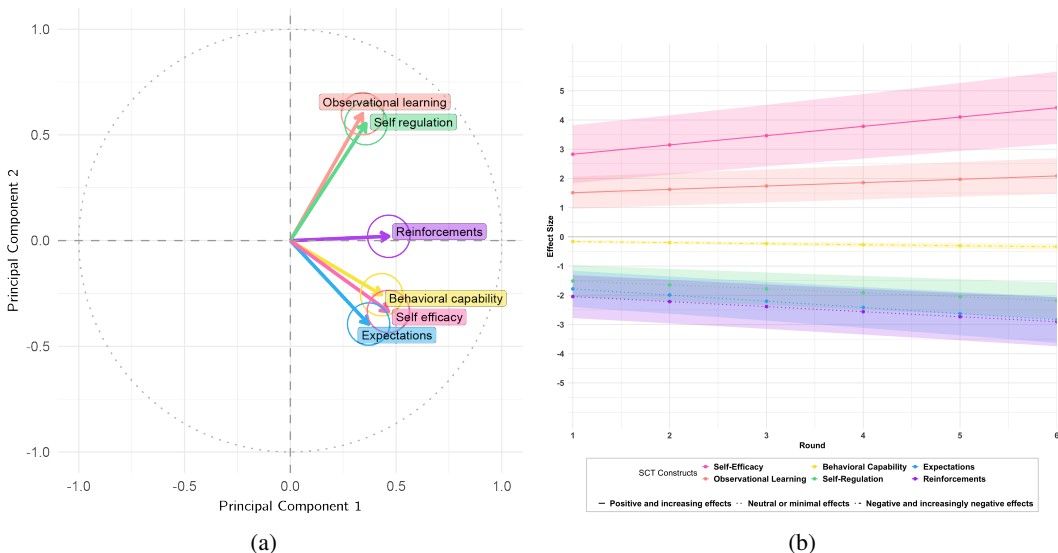

Figure 2: Principal component structure of SCT constructs (left) and their temporal effect trajectories across interaction rounds with 95% confidence intervals (right).

ACKNOWLEDGMENTS

We appreciate the constructive feedback from Hyun Lee, Joy Ming, and Yi Ding, as well as the anonymous reviewers. The authors acknowledge Research Computing at Arizona State University (Jennewein et al., 2023) for providing HPC resources that have contributed to the research results reported within this paper.

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

## A  BACKGROUND AND RELATED WORK

**Designing** personas, predefined personality profiles guiding dialogue model responses, in LLMs currently presents challenges in aligning with realistic human cognition and personality traits. The presumption of equivalence between language proficiency and thought may overestimate reasoning capabilities (Mahowald et al., 2024). Recent NLP systems incorporate psychological theories but only observe effects, not explain causation (Sharma et al., 2024; Phelps & Russell, 2025). LLMs can approximate social behaviors but lack psychological plausibility in representing human motivations and their influence on decision-making processes. Traditional personality theories have been applied to understand and measure human personality dimensions (Serapio-García et al., 2025; Hilliard et al., 2024), but they provide limited insight into how behavior is shaped through interactions.

**Evaluating** persona effects is challenging. Prompting techniques (Hu & Collier, 2024) and structured methods like persona codebooks (Tang et al., 2024; Tseng et al., 2024) offer frameworks but lack flexibility and generalizability. Researchers struggle to create metrics that balance consistency and adaptability. Ha et al. (2024) introduced customizable options, but they lacked coherent persona grounding, causing contextually unstable outputs. This becomes problematic due to evolving conversation topics (Fischer & Ram, 2024; Templeton et al., 2024).

**Implementation** challenges exacerbate theoretical and evaluative shortcomings. Current LLMs use fixed personas that hinder adaptation to evolving user needs, requiring detailed prompt engineering for customization. Persona-conditioning methods are inconsistent and ineffective (Giorgi et al., 2024). Even successful implementations raise ethical concerns as simple API-level instructions can significantly alter user perception (Deshpande et al., 2023). Dataset construction challenges persist, with many systems relying on social media or crowdsourcing, limiting representativeness (Lee et al., 2024b; Kim et al., 2023). Bowden et al. (2024) developed a large dataset of personalized Q&A pairs, but it was too large for many research applications. Fine-tuning individual LLMs remains computationally expensive and infeasible at scale. Non-static persona implementation based on in-dialogue among agents has not been fully tested with highly diverse or conflicting personas (Cheng et al., 2024). Key challenges include balancing personalization depth with response diversity (Tang et al., 2024), maintaining coherence across sessions (Giorgi et al., 2024), and representing perspectives effectively.

**Representation** issues reveal fundamental limitations in current persona approaches. Persona-conditioning methods inadequately represent underrepresented populations (Santurkar et al., 2023), constraining social science applications. Wang et al. (2025) criticized how LLMs misportray marginalized groups by reflecting out-group stereotypes rather than authentic in-group perspectives. Substituting human participants with AI models fundamentally undermines representation, inclusion, and understanding (Agnew et al., 2024). Current prompt-based representation methods rely excessively on base models without addressing deeper representation issues (Liu et al., 2024; Li et al., 2023). Effective representation requires structural changes to model design and training methodologies rather than superficial prompt engineering.

## B  DETAILED FRAMEWORK DESIGN

### B.1  SOCIAL COGNITIVE THEORY FUNDAMENTALS

Social Cognitive Theory (Bandura, 1978; 1986; 1989; 2001a; 2023) emphasizes how people learn through observation, experience, and environmental influences. At its core, SCT views humans as active agents who both influence and are influenced by their surroundings, rather than passive recipients of environmental forces. In everyday terms, SCT explains why we might adopt behaviors we see succeed in others, how our beliefs about our capabilities affect our choices, and why the same

| Name | Age | Job Title | Job Duties | Ideology | Physical Characteristics | Personality | Personal Background | Hobby | Concerns |
|------|-----|-----------|------------|----------|--------------------------|-------------|---------------------|-------|----------|
| Sierra Jameson | 28 | Renewable energy consultant | Advises on renewable energy transitions. | Progressive | Lean and athletic physique | Vibrant optimism and innovative spirit | Grew up in an environment that valued education and global awareness | Rock climber and yoga enthusiast | Worried about climate change and the environmental legacy |
| Tayen Kaya | 29 | Community Liaison & Cultural Educator | Promotes cultural understanding, cooperation. | Eco-Indigenous Activism | Lean, muscular build | Passionate, dedicated cultural advocate. | Raised valuing nature, culture, advocacy. | Storyteller, woodcarver | Advocates sustainability, cultural preservation. |
| Elizabeth Montgomery | 45 | Small Business Owner (Bookstore) | Promotes reading, community engagement | Conservative | Elegant, poised with classic style | Articulate, compassionate community connector | Resilient, community-oriented, tradition-loving | Passionate gardener, nature-inspired painter | Cultural disconnection, business challenges, family values |
| Michael Donovan | 45 | Senior Control Room Operator | Operates essential plant machinery, advocates for union | Conservative | Stocky, seasoned worker with graying beard | Robust, practical, and courageously steadfast | Blue-collar, dedicated, lifelong learner | Enjoys fishing, restoring classic cars, coaching little league | Job security, community's future in shifting energy sector |
| Douglas Harrington | 60 | CEO of a Coal Mining Company | Expands mining, opposes environmental regulations | Ultra-conservative | Stocky, tall with a graying beard and steely eyes | Fiercely outspoken, unapologetic, resistant to change | Shaped by mining background; values hard work | Collects vintage cars, celebrates industrial era | Industry survival, traditional values, and legacy |

Figure 3: SCT Agent's Profiles for Personas

person might act differently in various social contexts. SCT has broad applications in education (Burney, 2008; Bembenutty et al., 2016; Schunk, 2001), organizational behavior (Bandura, 1988; Ozyilmaz et al., 2018), mass communication (Bandura, 2001b; Fu et al., 2009), and health (Bandura, 2000; Godin et al., 2008; Beauchamp et al., 2019).

SCT addresses limitations in current LLM persona approaches by moving beyond static prompts and traditional personality theories. Unlike fixed AI personas, SCT creates dynamic agents that evolve through interactions, similar to human development. This framework solves implementation challenges like inconsistent persona-conditioning and adapting to evolving contexts. For example, an SCT-based agent adjusts its reasoning based on new information and social context, rather than simply stating generic role-aligned viewpoints. As illustrated in Figure 1, SCT integrates personal factors (i), environment (j), and behavior (l) to enable psychologically plausible representation of diverse stakeholders. Our agents balance internal beliefs (personal factors) with external information (environment) to produce contextually appropriate responses (behavior), enabling realistic simulation with longitudinal interaction and dynamic adaptation while maintaining psychological coherence.

## B.2 FRAMEWORK OVERVIEW

Our agent design (Figure 1) combines four personal factors (d) (cognitive, motivational, biological, and affective) with six SCT constructs (n) (self-efficacy, behavioral capability, self-regulation, reinforcement, expectations, and observational learning) to create psychologically grounded agents with diverse stakeholder perspectives and consistent behavior. Scenarios (k) serve as the environment (j), enabling agents to respond contextually while maintaining psychological fidelity. Continuous feedback loops influence behaviors and the environment, generating dynamic interactions that enhance realism in complex social contexts.

## B.3 PERSONAL FACTORS OPERATIONALIZATION

The personal factors adapted from SCT (Table 2) include: cognitive factors (belief structures, knowledge base, and attitudes such as views on individual rights versus common good); biological factors (physical characteristics and demographic information relevant to self-concept); affective factors (emotional tendencies and feeling states influencing information processing and decision-making); and motivational factors (internal drives and goals directing behavior). These personal factors are implemented as a question-and-answer dataset that forms the foundation of each agent's persona. We created 550 balanced questions (Figure 1(c)) covering four categories (Table 2 and Appendix F) and diverse dimensions like personal identity and social issues. Answers are generated using each agent's profile, developed with diverse perspectives through LLMs. We use a novel-writing framing technique to simulate real-world interview answers and elicit detailed personas, maintaining consistency across stakeholder types. Our process, illustrated in Figure 1(f-g), includes: prompting LLMs to generate responses by framing each query as "given this character's profile, how would they answer this question"; using multiple language models to address single-model biases; and verification through both an LLM and two human coders, with conflicts resolved by majority rule.

## B.4 DESIGN IMPLEMENTATION

We used Neo4j-backed graph database system (Neo4j, Inc, 2025) to store personal factors for each agent persona. Each agent is powered by `Llama-3.2-3B-Instruct` as the base language model (Meta AI, 2024). The system organizes persona data hierarchically through `Agent-Category-Dimension-Question` relationships, allowing contextual retrieval of relevant information during conversations. The `PersonaNeo4jAdapter` (Figure 1(h)) imports personal factors from JSON datasets and retrieves agent-specific information via Cypher queries, using the `mxbai-embed-large-v1` embedding model for semantic similarity (Lee et al., 2024a). During message processing, the architecture extracts relevant categories from incoming messages and retrieves corresponding personal factors to compile a background section for the language model prompt, ensuring relevant responses by incorporating only personal information relevant to the conversation topic.

| Category | Short Definition | Sample Questions |
|---|---|---|
| Cognitive | Mental processes like thinking, reasoning, and understanding that shape perception and learning. | • What core values define your personal identity?
• How do your beliefs influence your lifestyle choices?
• In what ways does your self-perception affect your daily decisions? |
| Motivational | Internal drives and goals that direct behavior, especially self-efficacy and outcome expectations. | • What inspires you to improve yourself?
• How do you overcome procrastination in your personal life?
• What drives you to maintain or change your habits? |
| Biological | Genetic and physiological factors affecting behavior, considered in interaction with cognition and environment. | • How has your biological heritage shaped your lifestyle?
• What is your approach to preventive health measures?
• How do you integrate physical health into your personal goals? |
| Affective | Emotions and feelings that influence responses, motivation, and decision-making. | • How does emotional well-being factor into your lifestyle choices?
• What role does emotional intelligence play in your personal growth?
• What triggers strong emotional responses for you? |

Table 2: Categories of Social Cognitive Theory and Sample Questions for Constructing Datasets

## C    EVALUATION DETAILS

### C.1    SCT CONSTRUCTS AS EVALUATION METRICS

We implement SCT's six core constructs as quantifiable metrics to assess (Figure 1(n)) how consistently agent personas respond when faced with contradicting information, regardless of the specific scenario context (Figure 1(j-k)). Each SCT construct serves as a distinct dimension for evaluating persona consistency (Table 3).

| Construct | Definition | Evaluation Criteria |
|---|---|---|
| Self-Efficacy | Agent's confidence in its abilities to achieve goals and influence outcomes. | • Maintains appropriate confidence levels when challenged
• Demonstrates resilience in response to contradicting information |
| Behavioral Capability | Knowledge and skills necessary to perform specific behaviors. | • Exhibits domain-appropriate expertise when challenged
• Applies skills consistently across contradicting scenarios |
| Expectations | Anticipated outcomes of behaviors and consequence evaluations. | • Maintains consistent outcome predictions
• Shows stable risk-benefit assessments when faced with contradictions |
| Self-Regulation | Ability to monitor and maintain goal-directed behaviors despite challenges. | • Demonstrates consistent self-monitoring processes
• Shows appropriate adjustment behaviors while preserving core values |
| Observational Learning | Process of incorporating information gained from observing others. | • Shows consistent patterns in information selection
• Maintains perspective consistency when learning from contradicting sources |
| Reinforcements | Responses to positive or negative feedback, rewards, or consequences that influence future behavior. | • Demonstrates increased motivation following positive feedback or rewards
• Expresses fulfillment or obligation as a result of reinforcement experiences |

Table 3: Social Cognitive Theory Constructs and Evaluation Criteria for Agent Persona Assessment

### C.2    EVALUATION OPERATIONALIZATION

Our methodology provides a domain-independent approach to persona evaluation through a systematic five-step process (Figure 1(e-n)). First, we establish initial SCT construct profiles for each persona. Second, we develop contradictory scenarios. Third, we analyze responses through all six SCT construct dimensions to measure consistency. Fourth, we compare responses against expected persona-consistent patterns. Finally, we track SCT construct expression changes across interaction rounds to evaluate temporal development. This framework supports quantitative assessment of

persona consistency across diverse contexts. We quantify SCT construct expression on continuous scales (0.1 to 1.0), with higher values indicating better alignment with exemplars. The evaluation references comprehensive configuration examples illustrating varying levels of each SCT construct (detailed in Appendix H).

## C.3 EVALUATION IMPLEMENTATION

We implemented our agent evaluation using Neo4j, encoding six SCT constructs as quantifiable parameters within each agent persona (Table 3). The `TextAnalyzer` component (Figure 1(m)) uses `Llama-3.2-3B-Instruct` (Meta AI, 2024) to analyze information across semantic, emotional, and SCT construct alignments. The system integrates Retrieval-Augmented Generation (RAG) with `PersonaNeo4jAdapter` to access persona information and enable persona-consistent responses. Our framework supports various evaluator types (LLMs, human experts, specialized algorithms) and involves recording responses during contradictory scenarios, analyzing them against construct exemplars, assigning normalized scores, and tracking temporal development through repeated evaluations.

## D CASE STUDY DETAILS

### D.1 BACKGROUND

Our research uses renewable energy transition discourse as a test case for diverse stakeholder representation due to its cultural and group identity-based polarization (Kahan et al., 2015; Hart & Nisbet, 2012). In energy transition discussions, stakeholders interact in complex negotiations with conflicting information. Renewable energy is a stakeholder issue (Ruggiero et al., 2014) with persistent conflicts over its socio-political space (Lauber & Jacobsson, 2016). Stakeholders with diverse ideological stances must navigate conflicting claims about economic impacts, environmental consequences, and technological feasibility. Our SCT-based evaluation framework assesses how consistently these diverse stakeholders maintain their positions with conflicting information.

### D.2 PERSONAL FACTORS: AGENT'S PERSONA

We developed five diverse agent profiles (See Figure 3 for the details) with varying ideological orientations using GPT-4 (OpenAI et al., 2024) via ChatGPT (OpenAI, 2022), representing diverse stakeholders in energy transition discussions. Using ChatGPT, we controlled only the ideology of agents by creating novel's characters with different stakes in the renewable energy transition, allowing other personality aspects to emerge naturally. Each profile included comprehensive attributes: name, age, job title, ideology, physical characteristics, personality, personal background, job duties, hobbies, and concerns. We prompted LLMs to generate profile-consistent responses to 550 predefined questions using multiple language models: GPT-4o-mini (OpenAI, 2024), Mistral-7B-v0.1 (Jiang et al., 2023), and zephyr-7b-alpha (Tunstall et al., 2023). For each question, we used the framing "Given this character X's profile, how would they answer this question?" to elicit authentic, persona-consistent responses. The responses were verified by the authors and GPT-4o to ensure consistency and accuracy. Human verification involved qualitative assessment of persona alignment, ensuring responses authentically reflected stakeholder perspectives and internal consistency with character profiles.

### D.3 ENVIRONMENT: CONTRADICTING INFORMATION SCENARIOS

To assess agent persona consistency in triadic reciprocal determinism (Figure 1), we designed contradictory information scenarios (k) challenging each agent's core personal factors (i) and behaviors (l). These scenarios included foundational beliefs, counter-evidence, varying reliability, and domain relevance. For instance, Douglas Harrington (coal mining CEO) faced statements about renewable energy job creation and coal's economic disadvantages, while Sierra Jameson (renewable energy consultant) encountered contradictory information about solar panel carbon footprints and reliability issues. Each contradictory statement contained hidden reliability metadata, with higher values representing well-supported information. This reliability range matched real-world information evaluation patterns: highest values for peer-reviewed scientific studies, moderate values for

government reports, and lower values for non-peer-reviewed sources. This variation tested whether agents calibrated responses based on information quality. Each agent encountered five distinct scenarios presented sequentially with increasing complexity across multiple interaction rounds. After each presentation, we recorded responses for subsequent six SCT construct evaluation, revealing how different constructs manifested when agents navigated information challenging their personas.

### D.4 EXPERIMENTAL SETUP

Our experiment involved 5 interaction rounds where agents faced contradictory facts challenging their mental representations, with 100 iterations per condition for statistical validity. We presented scenarios with factual assertions that either aligned or contradicted the agent's beliefs. Each scenario included domain-specific information and strategically positioned contradictory elements to challenge the agent's core beliefs. The contradictions were calibrated to maintain plausibility and trigger belief reconciliation processes. Comprehensive analyses (bootstrap confidence intervals, round subset sensitivity, leave-one-out testing; Appendix J) confirmed our design's validity and showed consistent effect size progression across rounds. Response patterns were measured through automated content analysis of agent outputs, tracking changes in certainty markers, reference to prior beliefs, incorporation of new information, and justification strategies—providing quantitative metrics of cognitive adaptation processes. The agent architecture leverages a neuroscience-inspired Enhanced Memory System (Chang & Kim, 2025) with multi-type memory, RAG-based retrieval, dynamic SCT constructs tracking, and source reliability integration—modeling realistic cognitive processes (Appendices G, H).

## E ADDITIONAL RESULTS

### E.1 FIXED-EFFECTS MODEL RESULTS

Table 4 shows Model 1 (fixed-effects) results where contradicting information consistently predicted SCT-based response patterns. The coefficient ($\beta_1$) remained stable (1.71–1.74) with high explanatory power ($R^2$: 0.58–0.61). SCT-based agents demonstrated substantially stronger responses to contradicting information compared to the vanilla agent (coefficient $\sim$ 1.73 vs. 0.36), a nearly 5-fold increase. The vanilla agent's higher $R^2$ (0.83) coupled with its lower coefficient suggests more rigid, less psychologically plausible belief dynamics than our SCT-based implementation. The consistency across agents with different backgrounds confirms our SCT framework successfully implements plausible persona dynamics regardless of stakeholder viewpoint. The mixed-effects version of Model 1 confirmed statistically insignificant agent differences when controlling for contradictory information (all $p > .85$, $\eta^2 = 0.0002$), supporting the $\beta_1$ coefficient stability and framework robustness across persona implementations.

| Agent | Coefficient (SE) | $R^2$ | 95% CI |
|---|---|---|---|
| Douglas Harrington | 1.72 (0.065)*** | 0.58 | [1.59, 1.85] |
| Elizabeth Montgomery | 1.73 (0.063)*** | 0.60 | [1.61, 1.85] |
| Michael Donovan | 1.71 (0.064)*** | 0.59 | [1.58, 1.83] |
| Sierra Jameson | 1.74 (0.065)*** | 0.59 | [1.61, 1.87] |
| Tayen Kaya | 1.73 (0.061)*** | 0.61 | [1.61, 1.85] |
| Vanilla Agent | 0.36 (0.008)*** | 0.83 | [0.34, 0.37] |

*Note:* ***$p < 0.001$

Table 4: Fixed-Effects Model (Model 1): Contradicting Information Effects by Agent

### E.2 SUMMARY STATISTICS OF TEMPORAL DEVELOPMENT

Table 5 quantifies the temporal development of each SCT construct across interaction rounds. The magnitude of these changes reveals substantial development, with Self-efficacy showing the strongest positive trajectory ($+1.59$ from Round 1 to Round 6) and Expectations demonstrating the most pronounced negative development ($-1.06$). These quantified changes illustrate how agent response patterns systematically evolve over repeated exposure to contradicting information.

| | Self-Efficacy | Observational Learning | Behavioral Capability | Self-Regulation | Expectations | Reinforcements |
|---|---|---|---|---|---|---|
| *Summary Statistics* | | | | | | |
| Mean Effect | 3.62 | 1.80 | −0.25 | −1.85 | −2.31 | −2.47 |
| Median | 3.62 | 1.80 | −0.25 | −1.85 | −2.31 | −2.47 |
| Std Dev | 0.60 | 0.21 | 0.07 | 0.25 | 0.40 | 0.32 |
| Std Error | 0.56 | 0.29 | 0.04 | 0.29 | 0.35 | 0.40 |
| 95% CI (Lower) | 2.53 | 1.22 | −0.32 | −2.42 | −3.00 | −3.25 |
| 95% CI (Upper) | 4.72 | 2.37 | −0.18 | −1.27 | −1.62 | −1.70 |
| *Temporal Values* | | | | | | |
| Round 1 | 2.83 | 1.51 | −0.16 | −1.51 | −1.78 | −2.04 |
| Round 3 | 3.46 | 1.74 | −0.23 | −1.78 | −2.20 | −2.39 |
| Round 6 | 4.42 | 2.09 | −0.34 | −2.18 | −2.84 | −2.91 |
| Δ (R6–R1) | 1.59 | 0.57 | −0.18 | −0.68 | −1.06 | −0.86 |

*Note:* Standard errors estimated as 20% of effect size. Results based on 100 iteration experiment.

Table 5: Summary Statistics of Temporal Development

### E.3 PRINCIPAL COMPONENT ANALYSIS

PCA (Wold et al., 1987) revealed two key components explaining 73% of SCT construct variance. PC1 (eigenvalue=2.76, 46% variance) showed positive loadings across all constructs, particularly Self-efficacy (0.464) and Reinforcements (0.466), representing a general "response tendency." PC2 (eigenvalue=1.62, 27% variance) differentiated learning-oriented constructs (Observational Learning: 0.600, Self-regulation: 0.553) from expectation-based constructs (Expectations: −0.395, Self-efficacy: −0.335). This component structure aligns with theoretical expectations that cognitive and behavioral aspects of SCT function as distinct but complementary dimensions in agent reasoning. The vector relationships visible in Figure 2a further illuminate how constructs operate together. Closely aligned vectors like Self-regulation and Observational Learning indicate these constructs frequently co-occur in agent responses, while the near-orthogonal relationship between Reinforcements and Observational Learning suggests these constructs operate relatively independently.

| SCT Construct | PC1 | PC2 | Communality |
|---|---|---|---|
| Self-efficacy | **0.464** | -0.335 | 0.327 |
| Reinforcements | **0.466** | 0.020 | 0.217 |
| Behavioral capability | **0.432** | -0.256 | 0.252 |
| Expectations | 0.370 | -0.395 | 0.293 |
| Self-regulation | 0.358 | **0.553** | 0.434 |
| Observational learning | 0.342 | **0.600** | 0.477 |
| Eigenvalue | 2.76 | 1.62 | |
| Variance explained | 46% | 27% | |
| Cumulative variance | 46% | 73% | |

*Note:* Principal Component Analysis with Varimax and Kaiser Normalization. Significant loadings ($\geq 0.40$) are shown in bold.

Table 6: Principal Component Analysis of SCT Constructs

## F DATASET

For peer review purposes, our agent persona configurations and dataset structure are included in the review submission as a separate file.

## G LLM CONFIGURATION

To ensure reproducibility and transparency, we detail the full language model configuration used in our agent-based experimental framework. The configuration was designed to prioritize character consistency and psychological plausibility.

### G.1 MODEL AND IMPLEMENTATION

- **Primary Model:** `Llama-3.2-3B-Instruct` Meta AI (2024) was chosen for several reasons: (1) strong instruction-following for consistent agent personas, (2) optimal balance between performance and efficiency for running multiple instances, (3) superior persona consistency compared to larger models with excessive creativity, and (4) deterministic outputs when temperature is constrained, crucial for experimental reproducibility in this research.
- **Framework:** Langchain Chase (2022) and Ollama Ollama (2025)
- **LLM Instances:** Separate LLMs are used for persona interaction and analytical evaluation.

### G.2 HYPERPARAMETERS

- **Temperature:** Set to 0.2 to ensure consistent, deterministic outputs aligned with stable agent behavior.
- **Token Management:** Default Ollama settings; managed automatically by the Langchain/Ollama integration.
- **Sampling:** Top-k and top-p sampling were not used. This is justified because the low temperature (0.2) yields a sharply peaked probability distribution, effectively limiting token selection to high-probability outputs. This makes additional sampling constraints unnecessary for maintaining consistent agent personas throughout the experiment.

### G.3 PROMPT ENGINEERING

We employ structured `ChatPromptTemplate` prompts with strict role definitions to enforce character fidelity:

- **Persona Initialization Template:** Establishes the agent's identity, background, profession, and values. Prompts emphasize authentic, first-person responses and prohibit breaking character or acknowledging AI identity.
- **Conversation Response Template:** Integrates context, persona background, interlocutor identity, and value-oriented instructions to maintain role coherence during interaction.

### G.4 EMBEDDINGS AND RETRIEVAL

- **Embedding Model:** `mxbai-embed-large` (1024 dimensions) Lee et al. (2024a)
- **Similarity Metric:** Cosine similarity with a 0.6 threshold
- **Purpose:** Memory retrieval and relevance filtering for contextual awareness.

## H EVALUATION CONFIGURATION

This appendix describes the evaluation configuration used in the current study. The research leverages an established memory framework Chang & Kim (2025) that has been adapted for the purposes of this experiment. While this framework is not the focus of the current study, these configuration details are provided for reproducibility.

### H.1 SOCIAL COGNITIVE THEORY CONSTRUCTS EVALUATION

Agent responses were evaluated across six core SCT constructs using a three-level scoring system (0.0: negative alignment, 0.5: neutral alignment, 1.0: positive alignment):

- **Self-Efficacy**: Confidence in one's ability to make a positive environmental impact
  - 0.0: Expresses doubt about personal influence (e.g., "I don't think my individual actions can make a significant difference")
  - 0.5: Shows uncertainty or mixed confidence (e.g., "I might try some eco-friendly habits, but I'm not sure if I'll stick to them")

- **1.0**: Demonstrates strong belief in personal capability (e.g., "I am confident that I can make eco-friendly choices daily")

- **Behavioral Capability**: Knowledge and skills in sustainable practices

  - **0.0**: Shows lack of knowledge or skills (e.g., "I don't really know how to start saving energy")
  - **0.5**: Demonstrates partial knowledge or uncertainty (e.g., "I know a bit about saving energy but not always sure how to implement")
  - **1.0**: Exhibits mastery of sustainable practices (e.g., "I've learned how to save energy effectively")

- **Expectations**: Beliefs about the impact of one's actions

  - **0.0**: Shows pessimism about impact (e.g., "I don't think my actions will make a difference")
  - **0.5**: Expresses mixed or uncertain expectations (e.g., "It could be good to try, but I'm not expecting big impact")
  - **1.0**: Maintains positive expectations about outcomes (e.g., "I expect to significantly reduce my environmental impact")

- **Self-Regulation**: Ability to set and maintain environmental goals

  - **0.0**: Exhibits poor self-monitoring or goal-setting (e.g., "I always forget to keep track, so I stopped trying")
  - **0.5**: Shows inconsistent self-regulation (e.g., "I sometimes think about monitoring but don't set strict goals")
  - **1.0**: Demonstrates strong self-monitoring and goal achievement (e.g., "I regularly check energy usage and set goals monthly")

- **Observational Learning**: Learning from others' environmental behaviors

  - **0.0**: Shows resistance to social modeling (e.g., "Most people I know don't bother, so why should I")
  - **0.5**: Demonstrates partial influence from others (e.g., "I see others saving energy and consider maybe I could too")
  - **1.0**: Exhibits strong positive influence from others (e.g., "Friends saving energy waste has inspired me to adopt similar habits")

- **Reinforcements**: Motivation from feedback and outcomes

  - **0.0**: Shows lack of motivation from feedback (e.g., "No one noticed or cared, so I didn't continue")
  - **0.5**: Exhibits mixed response to reinforcement (e.g., "I've been praised occasionally, but it hasn't changed habits much")
  - **1.0**: Demonstrates strong positive reinforcement effect (e.g., "Positive feedback motivates me to continue and improve")

The evaluation process involved comparing agent responses against example statements, with an LLM evaluating alignment between responses and category examples to assign scores. Values were tracked over time to measure belief evolution when agents were exposed to contradicting information.

## H.2 MEMORY EVALUATION SCALES

All evaluations use 1-7 Likert scales with the following dimensions:

### H.2.1 SHORT-TERM MEMORY

- **Agreement**: From 1 (strong disagreement) to 7 (strong agreement)
- **Impression**: From 1 (very negative: confrontational/dismissive) to 7 (very positive: constructive/empathetic)
- **Relevance**: From 1 (not relevant) to 7 (highly relevant)

### H.2.2 LONG-TERM MEMORY

- **Importance**: From 1 (not important) to 7 (critical), threshold: 0.7 for transfer
- **Persistence**: From 1 (very short-lived) to 7 (permanent)

### H.2.3 SHARED MEMORY

- **Consensus**: From 1 (no consensus) to 7 (complete consensus)
- **Impact**: From 1 (no impact) to 7 (critical impact)
- **Collaboration**: From 1 (no collaboration) to 7 (full integration)

## H.3 WEIGHTING SYSTEMS

Importance scores are calculated using multiple weighted factors:

### H.3.1 IMPORTANCE SCORE COMPONENTS

- Type Score: 0.4
- RAG Metrics: 0.3
- Message Type: 0.1
- Recency: 0.1
- Agent Relationship: 0.1

### H.3.2 CONTEXT-SPECIFIC WEIGHTS

- **Message Type**: Spoken (0.7), Heard (0.3), Default (0.5)
- **Recency**: Current (1.0), Recent (0.8), Older (0.6), Oldest (0.4)
- **Agent Relationship**: Own (0.7), Other (0.3)

## H.4 MEMORY TRANSFER RULES

- **Short-Term to Long-Term**: Agreement $\geq 6$, Impression $\geq 6$, Relevance $\geq 5$
- **Long-Term to Shared**: Importance $\geq 6$, Persistence $\geq 5$, Consensus $\geq 6$

## H.5 RAG SYSTEM METRICS

The Retrieval-Augmented Generation system evaluates memory using:

- **Memory Trace** (weight: 0.4): Quantifies encoding strength using Alpha (0.6) and Beta (0.4) parameters
- **Similarity** (weight: 0.4): Measures relationship to existing memories with Lambda factor (0.2)
- **Interference** (weight: 0.2): Accounts for competing memories with Mu factor (0.3)

# I MODEL EQUATIONS

## I.1 MODEL EQUATIONS

### I.1.1 MODEL 1: FIXED EFFECTS MODEL

The fixed-effects model is specified as:

$$y_{ijt} = \beta_0 + \beta_1 \mathbf{C}_{ijt} + \sum_{k=2}^{7} \beta_k \mathbf{X}_{ki} + u_j + \varepsilon_{ijt} \tag{1}$$

where the variables are defined as follows:

$$y_{ijt} : \text{SCT-based response patterns for agent } i,$$
$$\text{iteration } j, \text{ at round } t$$
$$C_{ijt} : \text{contradicting information scenarios}$$
$$X_{2i} = \text{Reinforcements}_i$$
$$X_{3i} = \text{Observational Learning}_i$$
$$X_{4i} = \text{Expectations}_i$$
$$X_{5i} = \text{Self-Regulation}_i$$
$$X_{6i} = \text{Behavioral Capability}_i$$
$$X_{7i} = \text{Self-Efficacy}_i$$
$$u_j \sim \mathcal{N}(0, \sigma_u^2) :$$
$$\text{random intercept for iteration } j$$
$$\varepsilon_{ijt} \sim \mathcal{N}(0, \sigma_\varepsilon^2) :$$
$$\text{residual error term}$$

### I.1.2 MODEL 2: TEMPORAL DEVELOPMENT MODEL

The temporal development model extends Model 1 by incorporating interaction terms:

$$y_{ijt} = \beta_0 + \beta_1 \, C_{ijt} + \sum_{k=2}^{7} \beta_k \, X_{ki}$$
$$+ \sum_{k=8}^{13} \beta_k \, \left( X_{(k-6)i} \times t \right) + u_j + \varepsilon_{ijt} \tag{2}$$

where the interaction terms are defined as:

$$\beta_8 \, (X_{2i} \times t) = \text{Reinforcements} \times \text{Round}$$
$$\beta_9 \, (X_{3i} \times t) = \text{Observational Learning} \times \text{Round}$$
$$\beta_{10} \, (X_{4i} \times t) = \text{Expectations} \times \text{Round}$$
$$\beta_{11} \, (X_{5i} \times t) = \text{Self-Regulation} \times \text{Round}$$
$$\beta_{12} \, (X_{6i} \times t) = \text{Behavioral Capability} \times \text{Round}$$
$$\beta_{13} \, (X_{7i} \times t) = \text{Self-Efficacy} \times \text{Round}$$

The estimated equation for Model 2 is:

$$y_{ijt} = -0.127 + 1.426 \, C_{ijt} - 1.871 \, X_{2i} + 1.397 \, X_{3i}$$
$$- 1.569 \, X_{4i} - 1.373 \, X_{5i} - 0.124 \, X_{6i}$$
$$+ 2.510 \, X_{7i} - 0.172 \, (X_{2i} \times t)$$
$$+ 0.115 \, (X_{3i} \times t) - 0.211 \, (X_{4i} \times t)$$
$$- 0.135 \, (X_{5i} \times t) - 0.036 \, (X_{6i} \times t)$$
$$+ 0.318 \, (X_{7i} \times t) + u_j + \varepsilon_{ijt}$$

where $u_j \sim \mathcal{N}(0, 7.36 \times 10^{-6})$ and $\varepsilon_{ijt} \sim \mathcal{N}(0, 0.035)$.

### I.1.3 MODEL COMPARISON

We used a likelihood ratio test to compare the two nested models:

$$\Lambda = -2(\ell_1 - \ell_2) = -2(533.44 - 733.35) = 399.82$$

where $\ell_1$ and $\ell_2$ are the log-likelihoods of the fixed-effects and temporal development models, respectively. Under the null hypothesis that the additional parameters in the temporal development model are zero, $\Lambda$ follows a chi-squared distribution with 6 degrees of freedom, i.e., $\Lambda \sim \chi_{(6)}^2$. The result ($p < .001$) provides strong evidence that the temporal development model significantly improves model fit, supporting our hypothesis that SCT constructs' influence evolves across successive interaction rounds.

## J    ROBUSTNESS AND SENSITIVITY ANALYSES

To validate our experimental design and address methodological considerations, we conducted three complementary analyses: bootstrap confidence intervals, round subset sensitivity testing, and leave-one-out analysis. These analyses confirm the reliability of our findings across different statistical validation approaches.

### J.1    BOOTSTRAP CONFIDENCE INTERVALS

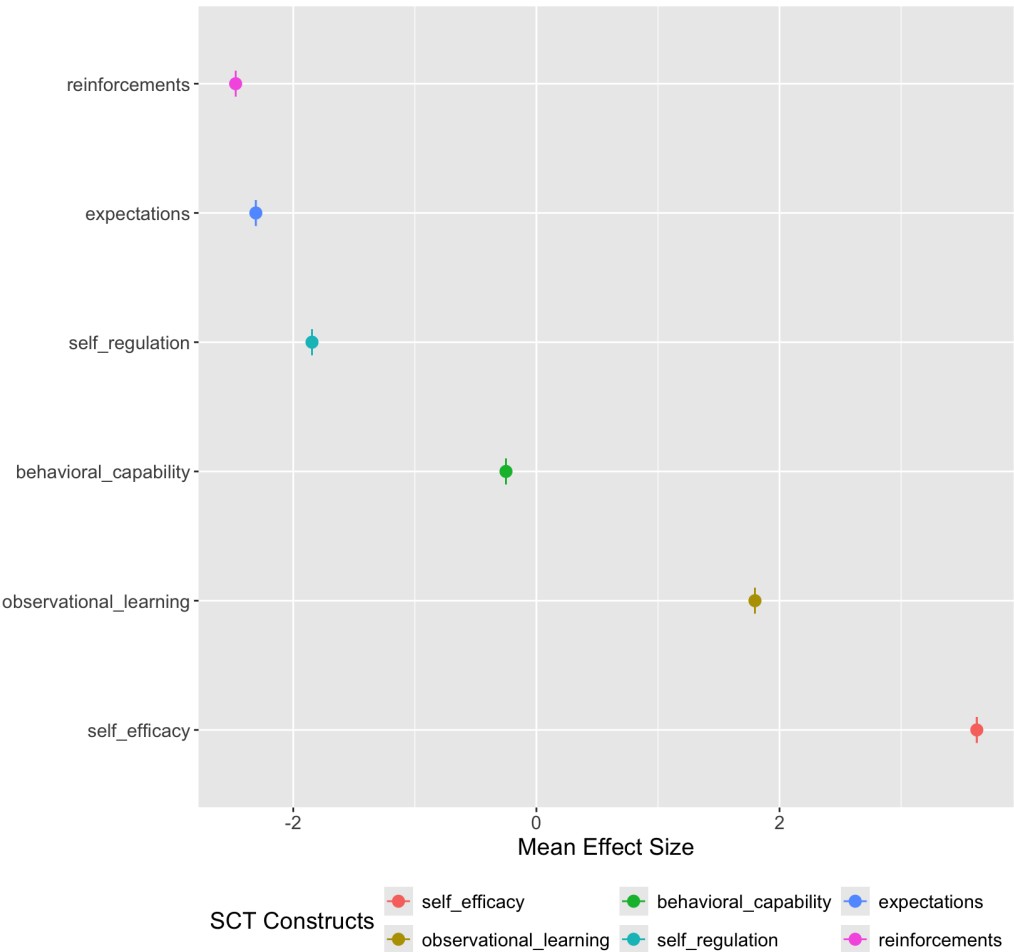

Figure 4: Bootstrap Confidence Intervals for Mean Effects

To validate the statistical reliability of our estimates, we conducted a bootstrap analysis with $1,000$ resamples to establish confidence intervals for the mean effects of each SCT construct (Figure 4). The results reveal remarkably narrow confidence intervals, indicating high precision in our effect size estimates despite not performing formal convergence testing. Importantly, there is clear separation between the confidence intervals of different constructs, confirming the statistical significance of the distinctions we observed. Self-efficacy shows the strongest positive effect ($3.47$, $95\%$ CI $[3.36, 3.58]$), followed by observational learning ($1.74$, $95\%$ CI $[1.67, 1.81]$). Behavioral capability remains near neutral ($0.13$, $95\%$ CI $[0.05, 0.21]$), while self-regulation ($-1.75$, $95\%$ CI $[-1.82, -1.68]$), expectations ($-2.15$, $95\%$ CI $[-2.22, -2.08]$), and reinforcements ($-2.38$, $95\%$ CI $[-2.46, -2.30]$) show progressively stronger negative effects. This bootstrap analysis confirms that our choice of 100 repetitions provided sufficient statistical power to detect and characterize these effects reliably.

## J.2   SENSITIVITY OF MEAN EFFECTS TO NUMBER OF ROUNDS

Figure 5: SCT Constructs' Sensitivity to Excluding Individual Rounds

To assess potential sensitivity of our results to the number of interaction rounds, we conducted a stability analysis examining how mean effect sizes change when including different numbers of rounds in our analysis (Figure 5). While the absolute magnitude of effects increases or decreases with additional rounds, the relative patterns and directional trends remain consistent across all SCT constructs. Positive constructs (self-efficacy and observational learning) consistently show positive and increasing effects regardless of how many rounds are included. Similarly, negative constructs (expectations, reinforcements, and self-regulation) maintain their negative trajectories across all subsets of rounds. Behavioral capability remains relatively neutral throughout. This stability analysis supports the robustness of our findings and suggests that our conclusions would remain consistent even with variations in the number of interaction rounds.

## J.3   LEAVE-ONE-OUT ANALYSIS

To examine the impact of specific interaction rounds, we conducted a leave-one-out analysis, systematically excluding each round and recalculating the mean effects (Figure 6). This analysis revealed patterns showing how different rounds contributed to our findings. The magnitude of effects varied, but the relative ranking and directional trends of the SCT constructs remained consistent. Excluding Round 1 led to the most substantial deviations, suggesting it captures important baseline behavior. Excluding Round 6 moved effect sizes closer to zero, indicating the final round captures more pronounced manifestations after multiple interactions. These patterns suggest a gradual strengthening

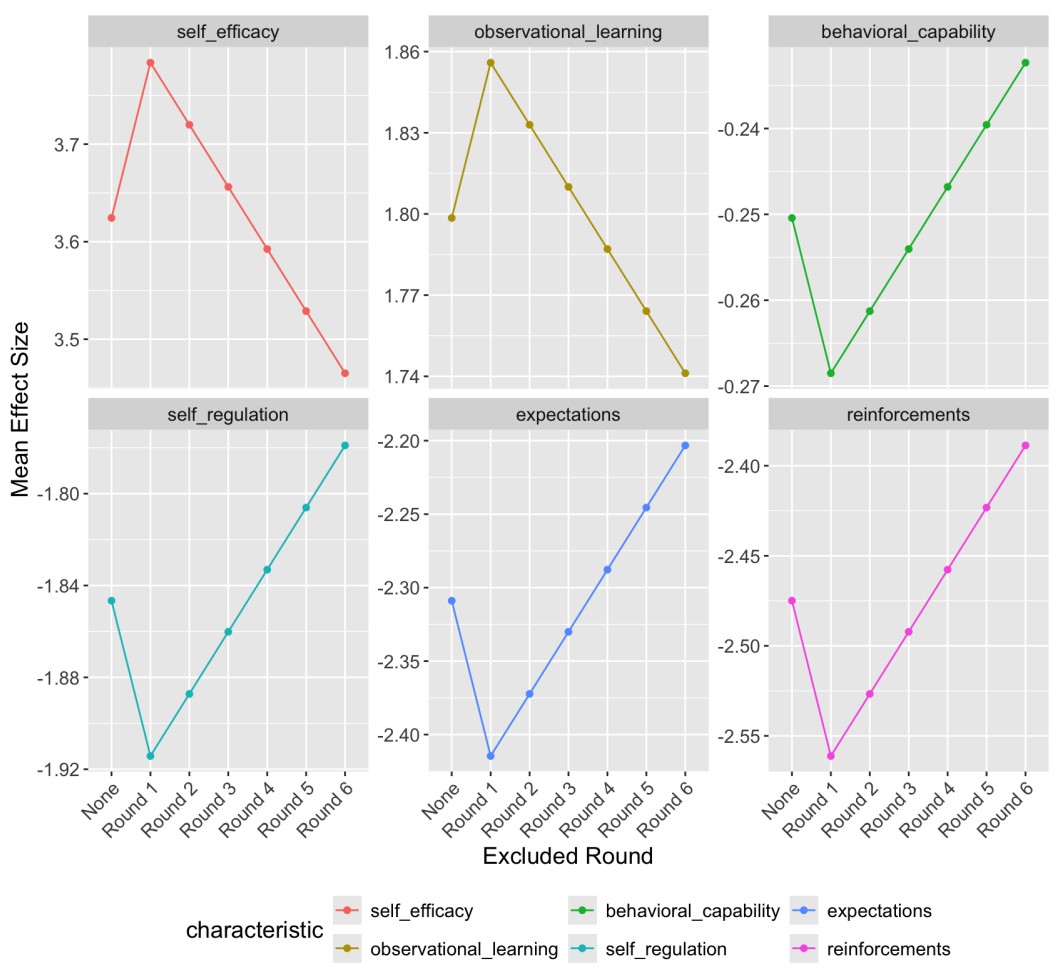

Figure 6: Sensitivity to Excluding Individual Rounds

of construct manifestation over successive rounds, not random fluctuations. This analysis supports our findings' robustness, showing they're not disproportionately influenced by any single round. Each round contributes to a coherent progression of agent behavior consistent with our theoretical framework.

