# OpenReview forum: "Persona Alchemy: Designing, Evaluating, and Implementing Psychologically-Grounded LLM Agents for Diverse Stakeholder Representation"
_ICLR.cc/2026/Workshop/AFAA — AFAA 2026 Poster_

### Official Review · Reviewer_PSMf · 2026-02-17
**Under-developed paper with some good ideas**

**Rating:** 2
**Confidence:** 3

**Summary:**

This paper introduces a framework for designing LLM agent personas grounded in Social Cognitive Theory (SCT), operationalizing it through four personal factors for design, six quantifiable constructs for evaluation, and a Neo4j graph database for implementation. Five diverse stakeholder agents in the renewable energy domain are tested against contradicting information of varying reliability across multiple interaction rounds. Results show consistent response patterns, systematic temporal evolution of SCT constructs (e.g., self-efficacy increasing, reinforcements decreasing), and a PCA structure explaining 73% of variance across two interpretable dimensions. The authors argue this provides more explainable and reproducible persona behavior than black-box prompting approaches, though the evaluation relies heavily on the same constructs used in design.

**Strengths:**

Grounding persona design in SCT rather than ad-hoc prompt engineering is a meaningful contribution. The operationalization of SCT's triadic reciprocal determinism into a concrete agent architecture is well-thought-out, and the paper does a decent job explaining why SCT is preferable to static personality trait theories like Big Five for dynamic agent behavior.

 The paper covers the full pipeline, design, evaluation, and implementation, rather than just one piece.  The bootstrap confidence intervals, round subset sensitivity, and leave-one-out analyses in Appendix E are thorough and demonstrate care about statistical validity.

I also like the choice of the case study: renewable energy transition is a genuinely polarized, multi-stakeholder domain that provides a good stress test for diverse persona representation.

**Weaknesses:**

The evaluation is fundamentally circular. The agents are designed using SCT constructs, the scenarios are crafted to test those constructs, and the evaluation metrics are those same constructs, scored by the same base model (Llama-3.2-3B). The paper essentially asks: "Do agents designed with SCT properties exhibit SCT properties when measured by SCT metrics?" The answer is almost tautologically yes. The R² values of 0.58–0.61 and the consistent coefficients across agents could simply reflect that the prompting pipeline reliably reproduces what was encoded, not that something psychologically meaningful is happening.

The paper claims to improve "diverse stakeholder representation" but never tests whether the agents actually represent stakeholders well. There is no study where domain experts (energy policy professionals, coal industry workers, environmental activists) evaluate whether these personas are realistic or useful. The human verification mentioned in Section 5.2 is limited to the authors checking consistency, this is not validation of representational quality.

The same model family serves as the persona agent and the analytical evaluator (TextAnalyzer). The paper should use a different model for evaluation, or better yet, include human evaluation.

Given these issues, I do not believe the paper is yet ready to be published.

---

### Official Review · Reviewer_mrPm · 2026-02-21
**This work grounds persona design in an established psychological theory.**

**Rating:** 4
**Confidence:** 4

**Summary:**

The paper's most valuable contribution is grounding persona design in an established psychological theory rather than ad hoc prompt engineering, which gives the evaluation metrics genuine theoretical justification. The use of a graph database for persona storage and the multi-model verification process for persona consistency both reflect thoughtful engineering choices. The temporal analysis across interaction rounds is genuinely interesting and adds a dynamic dimension absent from most persona work.

**Strengths:**

1) The work anchors persona design in SCT, giving a legitimate psychological foundation that most persona work with LLMs lacks.
2) Most persona research relies on qualitative or subjective assessments with no standardized measurement approach; this work converts SCT constructs into measurable, trackable scores across interaction rounds.
3) The examination of how SCT constructs evolve across interaction rounds adds a dynamic dimension that static persona frameworks miss. The finding that self-efficacy strengthens over time while expectations weaken is theoretically coherent and interesting.
4) The graph database architecture, detailed hyperparameter reporting, and multi-model verification process reflect genuine care for reproducibility, something many NLP papers underinvest in.
5) The bootstrap analysis, leave-one-out testing, and round subset sensitivity analysis demonstrate statistical rigor beyond what is typical for this type of work.

**Weaknesses:**

1) Personas are generated by LLMs and evaluated by LLMs using the same model family. There is no external human baseline to confirm that measured consistency reflects genuine psychological plausibility rather than model output artifacts.
2) Five agents in a single domain cannot support broad generalizability claims. The renewable energy context, while well-motivated, is ideologically specific and may not transfer to domains with different conflict structures.
3) Llama-3.2-3B-Instruct is a relatively small model. Claiming psychological plausibility while using a 3B parameter model raises questions about whether the observed behaviors reflect SCT dynamics or simply the constrained output space of a small model behaving predictably.
4) The 550-question persona dataset is generated using GPT-4 with limited justification for why these questions adequately capture the four SCT personal factors. The mapping from questions to psychological constructs is asserted rather than validated.
5) Human verification of persona consistency is mentioned, but no inter-rater reliability statistics are provided, making it impossible to assess the quality of the human coding process.

---

### Meta-Review · Area_Chair_2TwX · 2026-02-28

**Recommendation:** Main Papers Track
**Confidence:** 3

**Metareview:**

This paper proposes a framework for designing LLM agent personas based on Social Cognitive Theory. It turns the theory into clear, measurable factors and models them using a graph database. The approach is tested with several renewable energy stakeholder agents, tracking how their personas change over multiple interaction rounds.

Both reviewers appreciate the strong grounding in Social Cognitive Theory, the end-to-end system design, and the careful statistical analysis. They also find the renewable energy stakeholder setting a meaningful test case. However, Reviewer 1 notes limits in generalization (only five agents, one domain), the small base model, and the lack of human baselines or inter-rater reliability. Reviewer 2 raises a deeper concern that the evaluation may be circular, since the personas are designed, tested, and scored using the same SCT framework and model family.

While the paper is carefully built and grounded in theory, the main concern is that the evaluation may be circular. Because of this, the evidence does not clearly show that the agents truly reflect realistic stakeholders

---

### Decision · Program_Chairs · 2026-03-02

**Decision:**

Accept (Poster)

**Comment:**

The paper was originally submitted under the Main Track. While the paper is not ready to be accepted as a Main Track Paper, we find the work promising, and are giving an opportunity to the authors to instead get accepted in the Tiny/Short Track. Please submit a camera-ready version of up to 3 pages to comply with the Tiny/Short Paper requirements (more instructions on how to submit the camera-ready version will follow soon). The authors can also decide to withdraw if they prefer to not be accepted under the Tiny/Short Track.